

# Advancements in understanding and treating psoriasis: a comprehensive review of pathophysiology, diagnosis, and therapeutic approaches

Sai Chakith M. R.[1], Sushma Pradeep[2], Manu Gangadhar[1], Chaithra Maheshwari N.[3], Shuaib Pasha[4], Shiva Prasad Kollur[5], Nagashree S.[6], Chandan Shivamallu[4] and Satish Allur Mallanna[1]

[1] Department of Pharmacology, JSS Medical College, JSS Academy of Higher Education & Research, Mysuru, Karnataka, India
[2] Department of Radio Diagnosis, JSS Medical College, JSS Academy of Higher Education & Research, Mysuru, Karnataka, India
[3] Department of Microbiology, JSS Academy of Higher Education & Research, Mysuru, Karnataka, India
[4] Department of Biotechnology & Bioinformatics, JSS Academy of Higher Education & Research, Mysuru, Karnataka, India
[5] School of Physical Sciences, Amrita Vishwa Vidyapeetham, Mysuru, Karnataka, India
[6] Department of Information Science and Engineering, JSS Academy of Technical Education, Bangalore, Karnataka, India

Corresponding authors
Chandan Shivamallu,
chandans@jssuni.edu.in
Satish Allur Mallanna,
amsatish@jssuni.edu.in

## ABSTRACT

Psoriasis is a chronic autoimmune disease affecting millions worldwide. This condition is characterized by scaly, red patches of skin that can be painful, itchy, and disfiguring. This non-contagious illness forms plaques and accelerates the dermal cell's life cycle. This review provides a comprehensive overview of the current knowledge on psoriasis, covering its definition, prevalence, causes, pathogenesis, clinical features, diagnosis, and treatment options. The psychosocial impact of psoriasis on patients and their coping mechanisms is also explored. Biologic agents, which target specific cytokines involved in psoriasis pathogenesis, have revolutionized psoriasis treatment and have significantly improved patient outcomes. However, effective and safe treatments for moderate to severe psoriasis are still needed. Future research directions include the development of biomarkers for predicting disease severity and treatment response, investigating new therapeutic targets like the microbiome and epigenetics, and leveraging advancements in technology and genomics for deeper insights into psoriasis pathogenesis and treatment. This study summarizes the key aspects of psoriasis, including its epidemiology, pathophysiology, clinical traits, disease burden, and management. However, further research is needed to improve treatment outcomes and enhance the quality of life for patients affected by this complex condition.

## INTRODUCTION

A chronic, systemic immune-mediated inflammatory skin condition called psoriasis is characterized by red, scaly skin patches that can be unpleasant and itchy. It can occur at any age and affects between two and three percent of the world's population, though it most frequently strikes people between the ages of 15 and 35, it can also be found in very young children. The National Psoriasis Foundation states that psoriasis is a non-contagious disease that accelerates the life cycle of dermal cells, causing them to accumulate quickly on the skin's surface (*Bhosle et al., 2006*).

It is believed that a mix of hereditary and environmental factors contribute to psoriasis. In addition to certain triggers including stress, infections, and drugs, those with a family bloodline history of psoriasis are more prone to acquire the condition. Plaque psoriasis, which accounts for around 80–90% of instances of psoriasis, is the most prevalent form that disease can take. Other psoriasis kinds include erythrodermic, guttate, inverse, and pustular psoriasis, each with its own unique set of symptoms and traits. Psoriasis does not yet have a cure; however, several management and therapy alternatives can assist in managing its symptoms (*Falto-Aizpurua et al., 2020*). There are topical and systemic medications, with systemic options including oral, subcutaneous, and intravenous routes. Biologics are not the only subcutaneous medications; for example, methotrexate (MTX) can also be administered this way. Psoriasis can still significantly affect a person's quality of life and mental health even with current treatment options. Psoriasis patients should seek assistance from medical personnel and mental health specialists as required. Understanding one's disease, controlling flare-ups, and coping with the psychosocial difficulties of having psoriasis can all be greatly aided by counseling and education on the condition (*Hu et al., 2021*).

**Relevance to readers:** A wide range of readers, including dermatologists, scientists, and researchers with expertise in psoriasis research and therapy, are targeted by this review article. Additionally, it will help residents and medical students who need a thorough grasp of psoriasis for their education and future careers. This study contains important information that healthcare providers, including general practitioners, nurses, and other professionals involved in patient care, can use. Pharmacists and pharmacists with an interest in psoriasis treatment plans and pharmacological factors will also find this information beneficial. The article is also pertinent to biotech and pharmaceutical industry professionals working on new therapy development, as well as public health officials interested in the epidemiology and social implications of psoriasis. This evaluation will be helpful to academic institutions teaching and researching dermatology, patients with psoriasis and their advocates, lawmakers, and healthcare regulators who have an impact on treatment recommendations and healthcare policies for chronic illnesses. A lot of research is going on to find more evidence for the role and driving factors and shorten to the factors that already lead to the development of approved medications like JAK/TYK inhibitors. To enhance patient outcomes, direct future research, and influence clinical practice and policy, the article seeks to offer insightful information and updates on psoriasis.
## Search/survey strategy

Until April 2022, most databases were searched to find the most recent reported information on the definition, prevalence, causes, pathogenesis, clinical features, diagnosis, treatment options, and psychosocial impact of psoriasis. These databases included PubMed, Elsevier, Scopus, Google Scholar, and Web of Science. Psoriasis, prevalence, aetiology, pathogenesis, clinical features, diagnosis, treatment options, biologic drugs, cytokines, disease severity, response to treatment, microbiome, epigenetics, biomarkers, technology, genomics, epidemiology, and disease burden were among the keywords used in the search criteria (*Pradeep et al., 2025*). Every article received was examined and filtered to gather information about psoriasis that occurs naturally; unpublished findings and products from other sources were not included in this investigation.

## Prevalence and incidence

Psoriasis prevalence and incidence can differ by geographic location, age, and other variables. A total of 2.3% of the world's population had psoriasis, with Europe and North America having the highest prevalence rates. The study also discovered that the prevalence of psoriasis rose with aging, peaking in people 60 years and older. Additionally, it was discovered that men are more likely than women to have psoriasis (*Armstrong & Read, 2020*), The ratio of incidence rates between men and women is 85.5:73.2 per 100,000 person-years (*Pradeep et al., 2022b*).

According to a population-based study in 2015, the incidence of psoriasis in the United States is 98 per 100,000 person-years (*Rapp et al., 1999*).

Psoriasis incidence and prevalence vary significantly across regions, with only a few reliable studies on its incidence due to non-mandatory case registration. Incidence rates range from 0.60 per 1,000 person-years in Minnesota (1980–1983) to 10.36–15.04 per 1,000 in North African countries (2012). More studies have focused on prevalence, but comparing results is challenging due to differing methodologies (definitions, population ages, and diagnostic techniques).

Globally, prevalence tends to be higher in Northern European populations, such as Norway (up to 11.4%) and lower in Eastern Asia, with Taiwan recording a prevalence as low as 0.05%. Psoriasis affects adults more than children, with the highest prevalence seen in Western Europe, North America, and Australasia. Australia, Norway, and Israel have some of the highest lifetime physician-diagnosed prevalence rates. In 2017, it was estimated that 29.5 million people globally had psoriasis, representing a lifetime prevalence of 0.59% of the adult population (*World Health Organization, 2016*).

The WHO's global report on psoriasis also highlights the disparity in prevalence between high-income and low-income regions, with greater incidence reported in regions such as Europe and Australasia compared to East Asia. The prevalence in children is generally lower than in adults (*Sandoval, Pierce & Feldman, 2014*).

## Causes and risk factors

The exact cause of psoriasis is not fully understood, but research recommends that it is likely a combination of environmental and genetic factors. Genetic variations have been

identified that are related to an increased risk of developing psoriasis, particularly genes involved in the skin cell growth and immune system, according to the National Psoriasis Foundation. These genetic factors contribute to the underlying immune system dysfunction and abnormal skin cell proliferation observed in psoriasis (*Ritchlin, Colbert & Gladman, 2017*).

Alongside genetic factors, several environmental triggers have been identified that can contribute to the growth or worsening of psoriasis. Infections, particularly streptococcal infections, have been known to trigger psoriasis symptoms in some individuals. Certain medications, such as lithium (used for psychiatric conditions), antimalarial drugs, and beta-blockers (used for cardiovascular conditions), have also been associated with triggering or worsening psoriasis (*Pradeep et al., 2022b*).

Physical trauma to the skin, such as burns, cuts, or infections, can also act as triggers for psoriasis in susceptible individuals. Stress is another known trigger, with many people reporting that their symptoms worsen during periods of high stress. Lifestyle factors such as obesity and smoking have been related to an increased risk of expression of psoriasis as well (*Strober et al., 2016*).

It is important to note that while these triggers can contribute to the development or worsening of psoriasis, they do not affect every individual with the condition in the same way. Psoriasis is a multifactorial disease influenced by various factors, and the specific triggers and their impact can vary from one individual to another (*Takeshita et al., 2017*).

## Symptoms and diagnosis

Psoriasis can present with various symptoms depending on the type of psoriasis a person has. The most common symptoms include red, scaly patches of skin that are raised and have a silvery-white appearance in typical locations. These patches can be itchy or painful, and the skin affected by psoriasis may also be dry and cracked. Psoriasis can also affect the nails, causing them to thicken, develop pits, or develop ridges. Some individuals with psoriasis may also experience joint pain, stiffness, and swelling, which can be indicative of psoriatic arthritis, a condition that commonly accompanies psoriasis (*Menter et al., 2009*). It is important for the person experiencing these symptoms to look for medical attention for an accurate diagnosis and appropriate treatment.

Diagnosis of psoriasis is usually based on a physical examination of the skin and nails, as well as a review of a person's medical history and symptoms. In some cases, a skin biopsy may be done to confirm the diagnosis (*Bjerre et al., 2017*).

## Clinical features

Psoriasis is a persistent autoimmune condition that usually affects the skin, though it can also affect the joints and nails. Psoriasis patients experience varying rates of disease development; mild cases might persist for years in certain individuals, while moderate cases can escalate rapidly to moderate-to-severe cases. The progression of psoriasis is unpredictable, ranging from sporadic flare-ups to a persistently active condition. It may result in comorbidities such as metabolic syndrome, psoriatic arthritis, and an elevated risk of heart disease. Depression and anxiety are prevalent mental health conditions among

individuals with psoriasis. To further complicate its effects on general health, psoriasis frequently coexists with autoimmune conditions including rheumatoid arthritis and Crohn's disease. Depending on the type of psoriasis and the severity of the ailment, the symptoms and clinical characteristics of the condition can change (Table 1). Psoriatic arthritis is a form of arthritis that occasionally results from psoriasis, which can also affect the joints, we have knowledge of distinct types of psoriatic arthritis which react very differently to treatment (Parisi et al., 2015).

## Types of psoriasis

Psoriasis comes in a variety of forms, each with unique features (Fig. 1). Plaque psoriasis is the most prevalent and is distinguished by elevated, inflammatory skin lesions covered in thick, silvery scales. Small, drop-like lesions dispersed across the body are the first signs of guttate psoriasis. Skin folds, such as those in the groin and armpits, are impacted by inverse psoriasis (Kim, Jerome & Yeung, 2017). The appearance of pus-filled blisters distinguishes pustular psoriasis from erythrodermic psoriasis, which is characterized by widespread inflammation and scale shedding. Pitting, discoloration, and detachment from the nail bed are just a few of the changes that nail psoriasis can make to the texture and look of the nails, for those who suffer from this ailment, knowing the various forms of psoriasis aids in precise diagnosis and effective treatment planning (Gelfand et al., 2005).

### Plaque psoriasis

About 80–90% of instances of psoriasis fall into this category, making it the most prevalent kind. Raised, scaly, red patches of skin that are frequently covered with scales of silver give it away. The elbows, scalp, knees, and lower back are the areas of plaque psoriasis that are most typically affected. Plaque psoriasis is the most common type of psoriasis and has been extensively studied (Kamata & Tada, 2020). The following are some key findings on this type of psoriasis:

Plaque psoriasis has a significant genetic component, as evidenced by the identification of many genes associated with its onset. For example, a study discovered a substantial correlation between plaque psoriasis and a specific genetic variant in the CARD14 gene (Campione et al., 2020). Furthermore, the illness stems from an overactive immune system that produces inflammatory molecules known as cytokines. According to one study, cytokines including interleukin-17 and interleukin-23 were found to be more prevalent in patients with plaque psoriasis than in healthy controls (Brown, Claudio & Siebenlist, 2008). Additionally, several comorbidities, such as metabolic syndrome, psoriatic arthritis, and cardiovascular disease, are frequently present in conjunction with plaque psoriasis. Studies have indicated that compared to healthy people, persons with plaque psoriasis are more likely to develop metabolic syndrome (Armstrong et al., 2021; Boehncke & Schön, 2015).

### Guttate psoriasis

This psoriasis appears in kids, teenagers and adults after a bacterial illness like strep throat. Small, scaly, red spots on the skin that can arise unexpectedly and spread quickly across the body are its defining features (Gisondi et al., 2007).

**Table 1 Symptoms and clinical features of different types of psoriasis.**

| Sl. No | Types | Clinical features |
|---|---|---|
| 1 | Plaque psoriasis | Red, scaly patches of skin that can be itchy or painful |
| 2 | Guttate psoriasis | Scaly, small, teardrop-shaped, red or pink spots on the skin. |
| 3 | Inverse psoriasis | Smooth, red, and shiny patches of skin. |
| 4 | Pustular psoriasis | Pus-filled blisters that appear on the skin. |
| 5 | Erythrodermic psoriasis | Red, itchy, and painful skin |

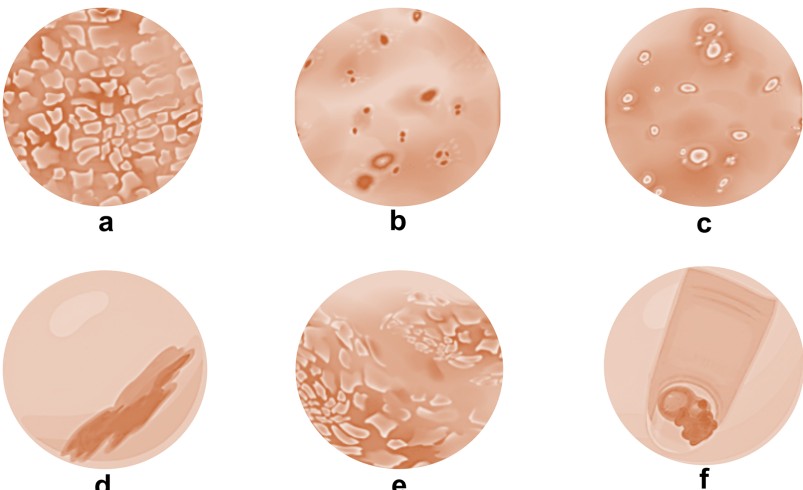

**Figure 1 Types of psoriatic skin conditions.** (A) Plaque psoriasis; (B) guttate psoriasis; (C) pustular psoriasis; (D) inverse psoriasis; (E) erythrodermic psoriasis and (F) nail psoriasis.

Guttate psoriasis is strongly linked to streptococcal infections, which are far more common in people with the condition than in healthy controls (*Kaur, Handa & Kumar, 1997*). Research has shown that some genes, including a variant in the IL23R gene, raise the likelihood of developing guttate psoriasis, even if the exact genetic basis of the disorder is still unclear (*Menter et al., 2010*). According to 41% of patients, stress is the main cause of guttate psoriasis, with other triggers including streptococcal infections, skin traumas, and some drugs (*Ohata et al., 2023*).

## Inverse psoriasis

Inverse psoriasis primarily affects skin folds such as the groin, armpits, and areas beneath the breasts. It is characterized by red, smooth, shiny areas that are often inflamed by friction and perspiration (*Leonardi et al., 2020*; *Gordon et al., 2018*). According to study, the most frequently affected locations are the groin (60%), axilla (40%), inframammary folds (25%), and intergluteal cleft (15%) (*Parisi et al., 2020*). Inverse psoriasis can be challenging to diagnose since it resembles other skin conditions such fungal infections and intertrigo; studies show that the average diagnostic delay for this condition is 4.4 years (*Pradeep et al., 2021*). Metabolic syndrome is more common in affected people than in

healthy people, and it is frequently associated with comorbid conditions including diabetes, obesity, and metabolic syndrome (*Ryan et al., 2014*).

### Pustular psoriasis

This is an uncommon variety of psoriasis that manifests as tiny, pus-filled blisters on the skin. Generalized pustular psoriasis (GPP), which can affect the entire body, and localized pustular psoriasis (LPP), which affects particular body parts like the palms and soles, are the two forms of pustular psoriasis (*Kaushik & Lebwohl, 2019*; *Gisondi, Del Giglio & Girolomoni, 2017*). With a prevalence of 0.1% to 0.5% in the general population, pustular psoriasis is extremely rare, accounting for less than 5% of all cases (*Bu et al., 2022*). LPP is more localized, with tiny blister patches on particular body areas, whereas GPP is characterized by painful blisters, fever, and chills (*Weber et al., 2021*). Streptococcal infections are commonly cited as a cause, and other common triggers include stress, infections, and several drugs (*Shivalingaiah et al., 2022a*).

The type and severity determine the prognosis. While LPP is typically simpler to manage with the right care, GPP can be fatal if left untreated. In comparison to individuals without pustular psoriasis, a study found that patients with GPP had an overall death rate of 3.6% (*Tsoi et al., 2012*).

### Erythrodermic psoriasis

Erythrodermic psoriasis is a rare but extreme form of psoriasis that can occur in individuals with pre-existing psoriasis or can be the first presentation of psoriasis. It is characterized by widespread red, inflamed skin that sheds scales in sheets. The skin may also be itchy, painful, and tender to the touch. Other symptoms can include fever, chills, rapid heartbeat, and fluid and electrolyte imbalances. It is a medical emergency that requires prompt diagnosis and treatment to prevent complications (*Zhou et al., 2022*).

A study reported that erythrodermic psoriasis affects less than 1% of individuals with psoriasis, but it can be life-threatening if left untreated. The authors noted that a variety of factors, including medication use, infection, and abrupt withdrawal of systemic steroids can trigger erythrodermic psoriasis. The diagnosis of erythrodermic psoriasis is based on clinical examination and skin biopsy, and treatment typically involves hospitalization, systemic medications, and supportive care (*Prignano et al., 2022*).

Another study emphasized the significance of early recognition and treatment of erythrodermic psoriasis to prevent complications such as infections, dehydration, and electrolyte imbalances. The initial management of erythrodermic psoriasis involves identifying and addressing any underlying triggers, such as infections or medication use, and providing supportive care. Systemic medications such as cyclosporine, methotrexate, and biological agents may also be used to reduce inflammation and improve symptoms (*Nestle, Kaplan & Barker, 2009*).

### Nail psoriasis

The nails are impacted by this type of psoriasis, which makes them thick, discolored, and pitted. The nails may potentially separate from the nail bed in extreme circumstances. It affects the nails, causing changes in their appearance and texture. It can occur in

individuals with other types of psoriasis or as an isolated condition. The most common nail changes seen in nail psoriasis include pitting, ridging, discoloration, and thickening. In severe cases, the nails may become deformed and detached from the nail bed, a condition known as onycholysis (*Kimball et al., 2008*).

A study reported that nail psoriasis affects up to 50% of individuals with psoriasis and is more common in those with psoriatic arthritis. The authors noted that nail psoriasis can significantly impact quality of life and can be challenging to treat due to the complex anatomy of the nail unit. Diagnosis of nail psoriasis is based on clinical examination and nail biopsy, and treatment may include topical or systemic medications, intralesional injections, and nail surgery.

Another study highlighted the importance of early recognition and treatment of nail psoriasis to prevent complications such as nail deformity and functional impairment. The authors noted that nail psoriasis can be difficult to diagnose and requires a multidisciplinary approach involving dermatologists, rheumatologists, and podiatrists. The authors also emphasized the importance of patient education and counseling to improve treatment adherence and improve outcomes (*Yorulmaz, 2023*).

Dermoscopy is a noninvasive, *in vivo* diagnostic procedure that has shown to be a useful tool in the assessment of nail psoriasis, a condition that is sometimes difficult to identify (*Falto-Aizpurua et al., 2020*). The conventional gold standard, nail biopsy, provides a conclusive diagnosis in around half of the cases; however, dermoscopy shows promise as a quick and trustworthy substitute. It is especially helpful in identifying early nail involvement because of its well-established diagnostic value in pigmented skin and nail illnesses. This may eliminate the need for biopsy in the diagnosis and follow-up of nail psoriasis patients. To facilitate diagnosis even before clinical indications of nail involvement become evident, dermoscopy serves as a crucial intermediary between clinical and histological tests (*Kamata & Tada, 2020*). When dermoscopic characteristics are present in clinically unaffected nails, it can act as an early indicator of disease activity, facilitating prompt treatment and intervention (*Shivalingaiah et al., 2022b*).

## Differential diagnosis

Clinical signs including red, scaly skin patches and other related symptoms are used to diagnose psoriasis. Consider alternative differential diagnosis, though, as several other skin disorders might mirror psoriasis (*Weber et al., 2021*). Lesions from discoid lupus might also resemble psoriasis. They often don't hurt or itch, though. Hair loss may result from scalp discoid lesions. These spots may remain on the skin for years without treatment. Individuals with darker skin tones could see the remnants of a discolored area that is either lighter or darker than the surrounding skin when discoid lupus resolves. Here are some of the common differential diagnoses of psoriasis (Table 2).

## Diagnostic tests

Psoriasis is typically determined based on clinical features, such as the appearance of the skin lesions and a history of the condition. However, in some cases, diagnostic tests may be needed to certify the diagnosis and exclude other conditions (*Parisi et al., 2015*).

**Table 2 Common differential diagnoses of psoriasis.**

| Sl. No | Skin disorders | Features |
|--------|----------------|----------|
| 1 | Seborrheic dermatitis | This is a common skin condition that affects the oily areas of the body such as scalp, and face. It is characterized by red, scaly, and itchy patches that can resemble psoriasis. |
| 2 | Eczema | Dry, itchy, and irritated skin are hallmarks of the chronic skin disorder eczema. It can resemble psoriasis, especially in children. |
| 3 | Pityriasis rosea | This is a self-limiting rash of skin that usually affects young adults. It is characterized by oval-shaped, pink, scaly patches on the trunk and limbs, and can resemble guttate psoriasis. |
| 4 | Cutaneous T-cell lymphoma | This is a rare type of skin cancer that can resemble psoriasis, especially erythrodermic psoriasis. |
| 5 | Fungal infections | Certain types of fungal infections, such as ringworm, can resemble psoriasis, especially in the early stages |
| 6 | Lichen planus | It is a skin condition characterized by itchy, flat-topped, purple-colored bumps. It can resemble psoriasis, especially inverse psoriasis. |

- Skin biopsy: A skin biopsy involves removing a small piece of skin tissue for inspection under a microscope. A skin biopsy can help authenticate the diagnosis of psoriasis and exclude other skin conditions that may mimic psoriasis (*Lebwohl et al., 2016*).
- Blood tests: This may be used to exclude other conditions, such as rheumatoid arthritis or lupus, that may be associated with psoriasis (*Puig, 2017*).
- X-rays and joint imaging: In some cases of psoriasis, imaging tests such as magnetic resonance imaging (MRI) or X-rays may be used to diagnose psoriatic arthritis, a type of arthritis associated with psoriasis (*Sandoval, Pierce & Feldman, 2014*).

High-resolution ultrasound (HRUS) has emerged as a superior tool for detecting early signs of inflammation in the joints and cartilage. HRUS allows for real-time visualization of soft tissues and can detect subtle inflammatory changes that may not yet be visible on X-rays. This makes HRUS a valuable diagnostic tool for catching psoriatic arthritis early, enabling prompt treatment to prevent long-term joint damage (*Sampogna et al., 2006*).

### Teledermatology

Teledermatology has revolutionised the treatment of psoriasis by improving access to care, particularly for people living in rural areas. Its advantages include making follow-ups and initial consultations easier by enabling patients to share pictures of their skin issues. This enables dermatologists to effectively track the evolution of their patients' diseases and suggest remedies. It has limitations when it comes to thorough assessments. For an accurate diagnosis, a complete inspection of the skin, including sensitive areas, is necessary, particularly when considering illnesses that may have stigmatizing elements connected with them. Traditional in-person consultations are still essential in this situation (*Parisi et al., 2015*).

When required, teledermatology should be used in conjunction with in-person examinations, according to guidelines from groups such as the American Academy of Dermatology, the German Dermatological Society, and the British Dermatological Society. Teledermatology increases productivity, cuts down on wait times, and eliminates travel problems, but to preserve patient confidence and the standard of care, data security and

diagnostic accuracy must be addressed. When treating psoriasis, balancing these methods can improve patient outcomes (*Pala, Bergler-Czop & Gwiżdż, 2020*).

## PATHOGENESIS OF PSORIASIS

Sustained inflammation that results in unchecked keratinocyte proliferation and defective differentiation is the defining feature of psoriasis. The inflammatory infiltrates of the psoriatic plaque are formed of dermal dendritic cells, macrophages, T cells, and neutrophils. They are overlaid by acanthosis (epidermolysis), according to the histology of the condition (Fig. 2). The inflammatory mechanisms involved in plaque psoriasis and the other clinical variants are similar, but they also show distinct distinctions that are responsible for the variations in phenotype and therapeutic response (*Satish et al., 2023*).

Psoriasis pathogenesis is a complex process with several interrelated components. In afflicted skin regions, it starts with the aberrant behavior of keratinocytes, which causes fast proliferation and improper differentiation (*Tan, Chong & Tey, 2012*). Cytokines and their receptors impact this process by initiating complex signaling pathways including MAPK, STAT, and NF-κB, which intensify the inflammatory response. In addition, the significance of the microbiome has become more and more clear, with dysbiosis playing a part in the imbalance. Psoriatic lesions persist because of pro-inflammatory conditions that are created by microbial changes, immune system disruptions and disruptions in cell signaling pathways. The chronic inflammation and aberrant skin cell proliferation associated with psoriasis are caused by the ongoing interaction of keratinocytes, cytokine signaling, and the microbiota. This highlights the disease's complexity and the requirement for focused therapy approaches (*Ippagunta et al., 2016*).

### The role of keratinocytes in psoriasis pathogenesis

Both the initial stages of psoriasis and its ongoing maintenance depend heavily on keratinocytes. Keratinocytes have a variety of stimuli they might react to as part of the innate immune system. The actions of keratinocytes are modulated, and psoriasis is influenced by a variety of variables including genetics, cytokines and receptors, metabolism, cell signaling, transcription factors, non-coding RNAs, antimicrobial peptides, *etc.* Self-nucleotides and antimicrobial peptides are released by stressed keratinocytes, which aid in pDC activation. Then, IFN-α, IFN-γ, TNF-α, and IL-1β are produced by the activated and matured mDCs (*Wang et al., 2020*).

In addition to taking part in the initiation phase, keratinocytes also contribute to the maintenance phase by amplifying psoriatic inflammation. Once activated by proinflammatory cytokines synergistically, keratinocytes are highly proliferative. They can produce copious chemokines (*e.g.*, CXCL1/2/3, CXCL8, CXCL9/10/11, CCL2, and CCL20) to recruit leukocytes (such as neutrophils, Th17 cells, dendritic cells, and macrophages), antimicrobial peptides (*e.g.*, S100A7/8/9/12, hBD2, and LL37) to induce innate immunity, and other inflammatory mediators to amplify inflammation. Additionally, keratinocytes, fibroblasts, and endothelial cells cause tissue reorganization by promoting the growth and activation of endothelial cells as well as the deposition of extracellular matrix. Keratinocyte hyperproliferation and aberrant differentiation, dilated and hyperplastic blood vessels, and
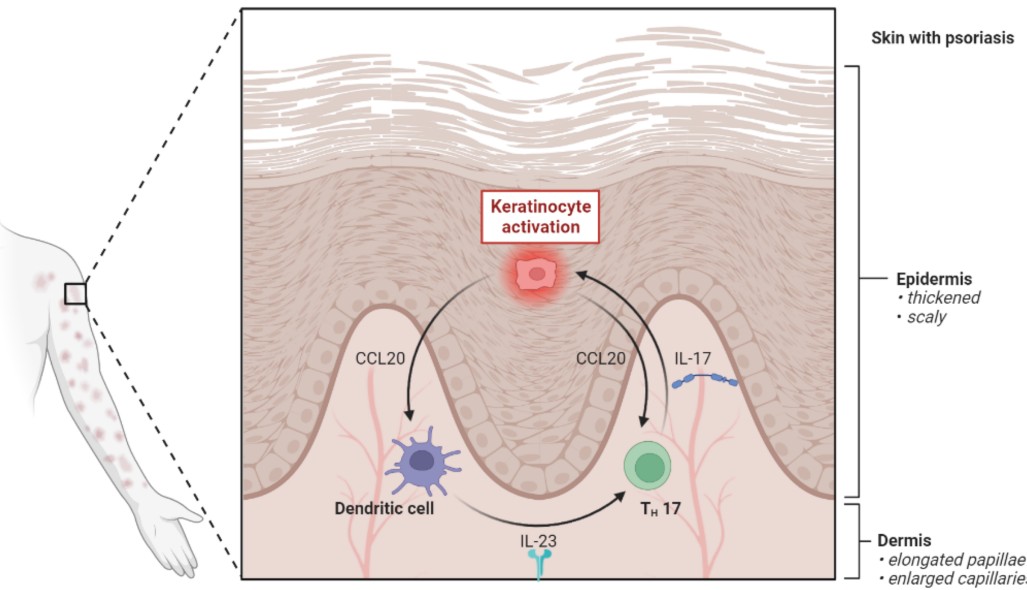

**Figure 2 Pathogenesis of psoriasis.** The pathogenic axis of IL-23 and IL-17 is what causes psoriasis. The maturation of myeloid dendritic cells (mDCs) and their production of TNF-, IL-12, and IL-23 are encouraged by the activation of plasmacytoid dendritic (pDCs), which in turn triggers the activation of Th (T helper) 1 and Th17 and the release of inflammatory cytokines including TNF-, IL-17, IL-21, and IL-22. When these cytokines, particularly IL-17, activate keratinocytes, they create antimicrobial peptides, cytokines, and chemokines that increase inflammation.

infiltration of inflammatory cells such as leukocytes are all consequences of the interaction between keratinocytes and immune cells, particularly Th17 cells, which leads to the production and maintenance of psoriasis (*Pradeep et al., 2022a*). This process is explained below in Fig. 3.

## The role of cytokines and their receptors in the pathogenesis of psoriasis

The development of psoriasis depends on cytokines and their receptors, which allow immune cells and keratinocytes to communicate. TNF-$\alpha$, IFN-$\gamma$, IL-23/IL-17A, and IL-22 are immune-derived chemicals that stimulate keratinocytes, which set off several signaling pathways. Antimicrobial proteins, cytokines, chemokines, and growth factors are released due to aberrant keratinocyte development. To treat psoriasis, therapies targeting TNF-$\alpha$, IL-17A, and IL-23 specifically work very well (*Zhou et al., 2022*). Researchers' focus has recently increased on cytokines produced by keratinocytes or receptors expressed in those cells (*Micali et al., 2019*).

Psoriasis is thought to be primarily caused by the IL-23/IL-17 cytokine axis. Immune cells are thought to need to produce IL-23 to maintain and grow immune cells capable of generating IL-17. However, keratinocytes can also generate IL-23, however it is unknown if this IL-23 plays a part in psoriasis. Scientists recently showed, using a transgenic mouse model, that keratinocyte-derived IL-23 was sufficient to activate IL-17-producing immune cells, induce them to release IL-17, and result in a persistent skin inflammatory response. Further research revealed that IL-23 expression in keratinocytes was controlled by

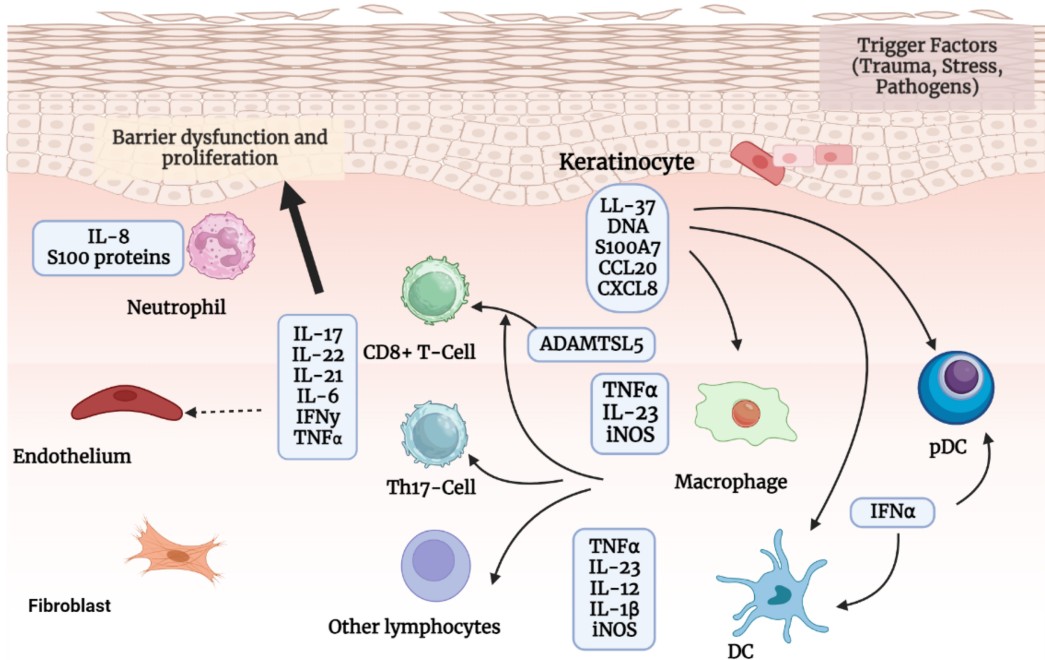

**Figure 3 The pathogenic process of psoriasis.** The pathogenic process of psoriasis is mostly shown in this image from the viewpoint of keratinocytes. Initial stimuli can excite keratinocytesand stressed keratinocytes produce self-nucleotides and antimicrobial peptides, activate pDCs and later DCs, and participate in psoriasis beginning phase. Activated keratinocytes impact the pathogenesis of psoriasis after being stimulated by cytokines in ways such as inflammatory infiltration, epidermal hyperplasia, innate immunity, tissue reorganization, *etc.*

epigenetic regulation through H3K9 dimethylation, which may contribute to psoriasis (*Kimball et al., 2008*).

Another important cytokine generated mostly by CD4+ T cells and group 3 innate lymphoid cells (ILC3) after IL-23 is IL-22. Non-hematopoietic cells including keratinocytes, epithelial cells, and hepatocytes express its receptor, known as IL-22R. By preventing keratinocytes from differentiating to their final state, IL-22 promotes the development of psoriasis and promotes the production of proinflammatory chemokines and antimicrobial peptides. IL-22 binding protein (IL-22BP) is a naturally occurring IL-22 inhibitor that binds specifically to IL-22, preventing it from performing its biological role. elevated levels of epidermal thickening and elevated production of inflammatory cytokines and IL-22-inducible antimicrobial peptides were seen in IMQ-induced psoriasis-like skin condition, which was aggravated by both hereditary IL-22BP deficiency and anti-IL-22BP neutralizing antibody (*Gisondi et al., 2021*).

TNF-like weak inducer of apoptosis (TWEAK), a crucial cytokine in psoriasis, has recently come to light. TWEAK deficiency reduced psoriatic dermatitis brought on by IMQ. Furthermore, animals lacking fibroblast growth factor-inducible 14 (Fn14, the TWEAK receptor) also had less severe illness. Scientists have most recently shown that keratinocytes play a crucial role in TWEAK's function in psoriasis utilizing a mouse model with keratinocyte-specific deletion of Fn14. Mice were shielded against IMQ-induced inflammation and psoriasiform hyperplasia by Fn14 loss in keratinocytes. It is significant

to note that inhibiting TWEAK had a comparable effect on reducing epidermal thickness, skin infiltrates, and inflammatory mediators as inhibiting TNF-α and IL-17A (*Chauhan et al., 2020*).

## Role of microbiome in psoriasis pathogenesis

Research on the impact of the microbiome on psoriasis severity is emerging, providing insight into the complex interplay between the body's microbial ecosystems and an inflammatory, chronic skin disorder. Dysbiosis, which is marked by changes in the variety and makeup of the microbiota and includes opportunistic pathogens such as human papillomavirus, *Malassezia*, and *Candida albicans*, as well as bacteria (*Staphylococcus aureus* and *Streptococcus pyogenes*). Psoriasis and dysbiosis are closely related conditions (*Baliwag, Barnes & Johnston, 2015*).

### Psoriasis and skin microbiome

Changes in the skin microbiome are crucial for psoriasis. Disturbances in the makeup and operation of the microbial communities that live on the skin are associated with psoriatic flares. Because of these microbial imbalances, psoriasis, which is characterized by reduced barrier function, displays disruptions in the integrity of the epidermal barrier (*Chen et al., 2020*). The bacterial and fungal flora of the skin, such as *Corynebacterium* species and *Candida albicans*, contribute to the build-up of Th17 cells, which exacerbates inflammatory reactions. The skin barrier structure is maintained by the skin's resident microbiota and epidermal immune cells. Pathological circumstances can disrupt this structure, which in turn can exacerbate inflammatory reactions in psoriasis (*Parisi et al., 2015*).

### Psoriasis and gut microbiome

The regulation of systemic immunity by the gut microbiota is a critical factor in the pathogenesis of psoriasis. Increased gut barrier permeability brought on by dysbiosis in the gut microbiome can facilitate the translocation of microbial antigens and their metabolites into circulation. This process contributes to the onset and progression of psoriasis by inducing systemic and local immune responses. Studies have indicated that psoriasis is associated with malfunctioning intestinal barriers that upset the immunological system's and microbiota's delicate balance (*Chen et al., 2020*). The relationship between gut dysbiosis and the severity of psoriasis is further supported by the elevated amounts of markers such as claudin-3 and intestinal fatty acid binding protein (I-FABP) in psoriasis patients, which indicate a weakened intestinal barrier (*Farley & Menter, 2011*).

The gut and skin microbiomes have a major impact on how severe psoriasis is. The integrity of the barrier may be compromised by dysbiosis and disturbances in microbial populations, which can exacerbate psoriatic symptoms and trigger immunological reactions. Comprehending these intricate relationships presents prospective paths for focused treatments and measures, intending to adjust the microbiota and reduce the intensity of psoriasis (*Parisi et al., 2015*).

## Cell signaling

The signaling pathway, which affects both keratinocytes and immune cells, is a key regulator of psoriasis and is implicated in a variety of biological factors. Nuclear factor-kappa B (NF-κB), Signal transducer and activator of transcription (STAT), MAPK, and other significant signalling pathways are changed in psoriasis (*Megna et al., 2020*).

### NF-κB signaling pathway

NF-κB signaling affects immune cells and keratinocytes, which has been demonstrated to contribute to the development of psoriasis. In psoriatic patients' lesional skin, NF-κB was intensely active. Various mice models have shown that aberrant NF-κB activation in keratinocytes and T cells is crucial for the emergence of inflammatory skin conditions like psoriasis. Skin symptoms resembling psoriasis appeared in mice with IκBα global deletion, although IκBα loss in keratinocytes only produced epidermal hyperplasia without epidermal inflammation. However, the loss of IκBα in keratinocytes and T cells produced a phenotype that was comparable to that of global deficit. Furthermore, the phenotype that emerged in global IκBα knockout animals was reversed by the targeted deletion of RelA from keratinocytes. Additionally, Imiquimod-induced psoriasis-like dermatitis was less severe in mice lacking in NF-κB. These findings suggest that NF-κB activation is necessary for psoriasis to develop in immune cells as well as keratinocytes (*Pradeep et al., 2022a*).

The phosphorylation, ubiquitination, and proteasomal degradation of the IκBs occur as a result of the stimulation of several cell membrane receptors, including the tumour necrosis factor receptor (TNF)R, IL-1 receptor, Toll-like receptor, T-cell receptor (TCR), and B-cell receptor (BCR). IκB kinases (IKKs), α and β which are complexed with the regulatory component NEMO (NF-κB essential modulator; IKKγ), catalyse the phosphorylation at two serines at the amino-terminus of IκB. The active IKK complex primarily uses IKKβ to phosphorylate IκB. This causes adjacent lysine residues to undergo lysine 48 (K48)-linked polyubiquitination, which is started by the ubiquitin E3 ligase complex Skp1/Cul1/F-box protei–TrCp. The NF-κB bound IκB is then degraded at the 26S proteasome because of this. Once in the nucleus, free NF-κB dimers link to NF-κB DNA sites and start the transcription of genes (Fig. 4) (*Rosenbach et al., 2010*).

### STAT signaling pathways

It is well established that the JAK (Janus kinase)/STAT signaling is crucial for the development of psoriasis. Notably, STAT3 is the most hyperactive of the STATs and controls cell proliferation, differentiation, and death in both immune cells and keratinocytes. In response to several inflammatory cytokines in psoriasis, such as IL-6, IL-17, IL-21, IL-19, IL-22, *etc.*, STAT3 plays a key function in keratinocytes. Inhibition of cell differentiation, increased cell proliferation, and generation of antimicrobial peptides were all effects of STAT3 activation in keratinocytes. The psoriasis-like lesions with cytokine profiles comparable to those of human psoriatic plaques spontaneously developed in the transgenic mouse model overexpressing STAT3 in keratinocytes. Additionally, selective ablation of STAT3 in keratinocytes as opposed to T cells decreased dermatitis like

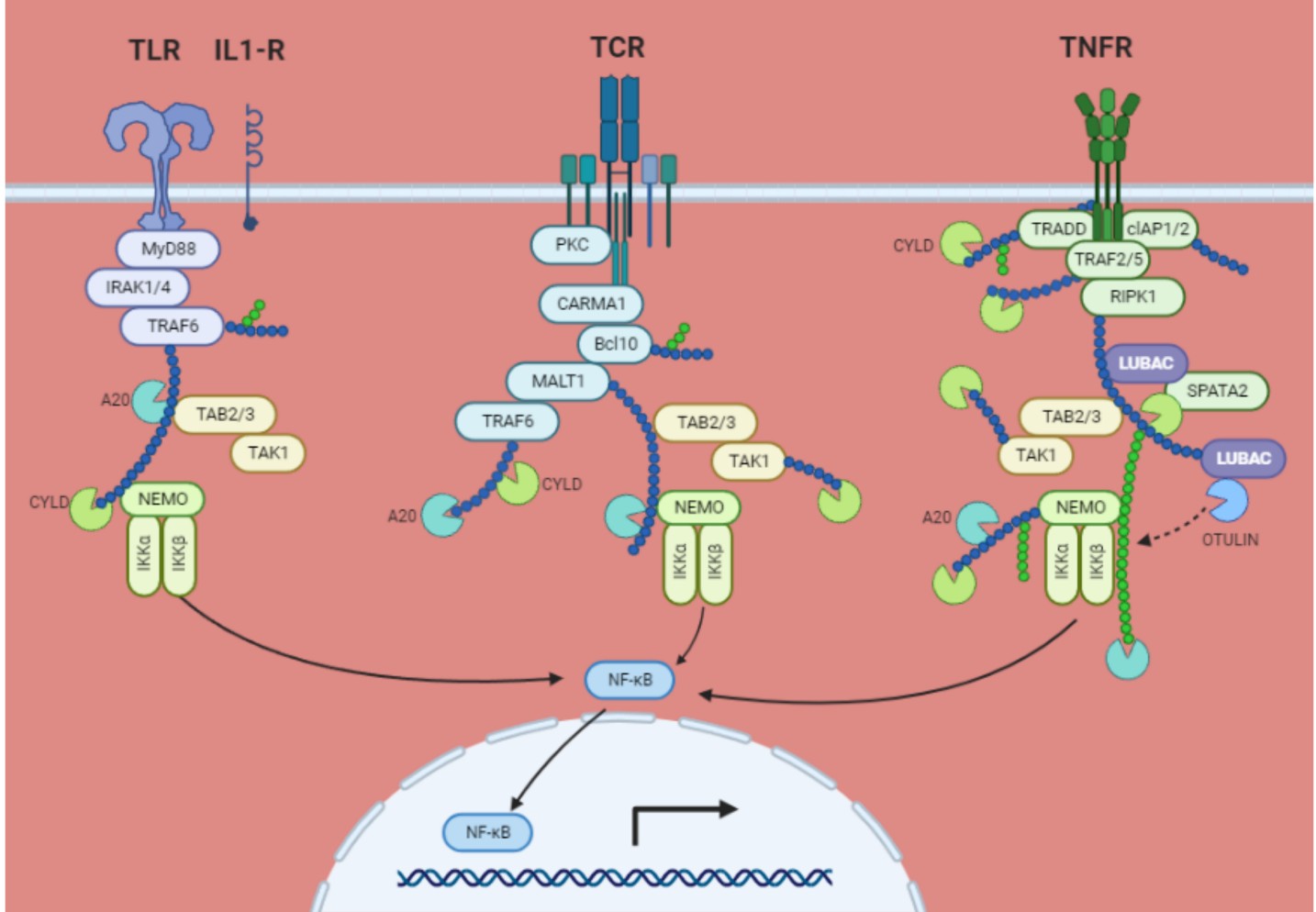

**Figure 4 NF-κB activation *via* the IκB degradation route is a common mechanism.** TRAF2, TRAF6, RIP, MALT1, and NEMO are all K63 polyubiquitinated in response to ligand interaction of certain membrane receptors. Through their interaction with TAB2 and TAB3, the poly-ubiquitin chains attract the TAK kinase complex. Activated TAK1 may phosphorylate and activate IKKβ, which in turn phosphorylates IκB bound to cytosolic NF-κB, causing its destruction by the proteasome and the βTrCP E3 ubiquitin ligase. The target genes are subsequently transactivated when free NF-κB moves to the nucleus. By removing K63 ubiquitinated chains from active TRAFs, RIP, and NEMO, deubiquitinating enzymes CYLD and A20 may prevent NF-κB activation. A20 may also stop NF-κB activation brought on by TNF-α by catalyzing the K48 ubiquitination of RIP, which results in its proteasomal destruction. The TNF receptor (TNFR1) not only encourages survival by activating NF-κB target genes, but it also increases antagonistic apoptotic pathways.

psoriasis. Therefore, STAT3 in keratinocytes plays a more significant role in the development of psoriasis (*Shivalingaiah et al., 2022a*) this is shown in Fig. 5.

### *MAPK signaling pathway*

In addition to being crucial in controlling keratinocyte proliferation and immune response, MAPK kinases are implicated in the pathogenesis of psoriasis. P38 was active in the psoriatic epidermis, and cutaneous stimulation of p38 led to dermatitis in mice that resembled psoriasis. Dermatitis brought on by IMQ was reduced by topical p38 inhibitor treatment. Studies conducted *in vitro* demonstrated that p38 inhibitors reduced

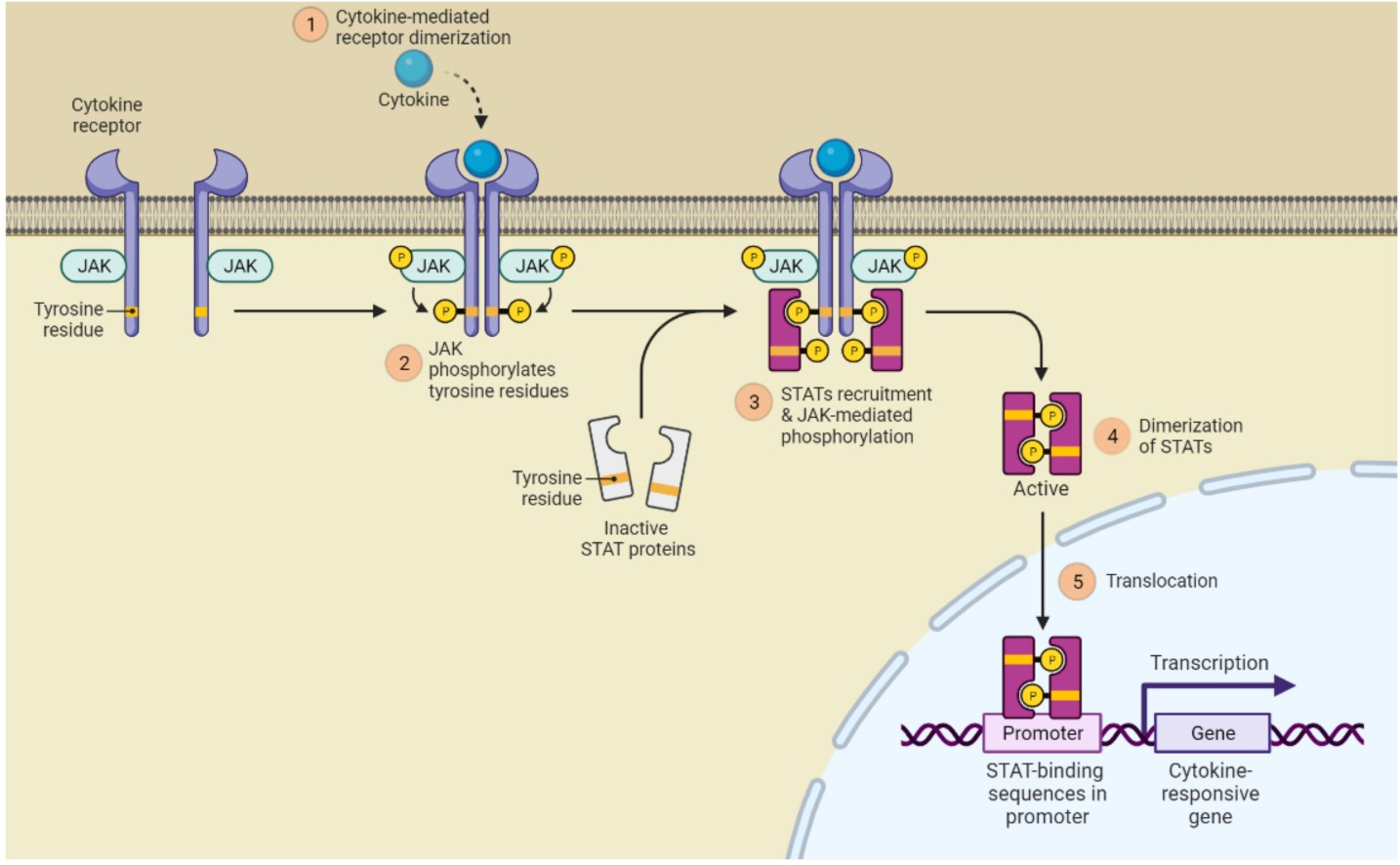

**Figure 5** **JAK/STAT signaling pathway activation.** (1) Cytokines and growth factors bind to the appropriate receptors, which causes the receptor to dimerize and draw in associated JAKs; (2) Tyrosine phosphorylation of the receptors and the creation of STAT docking sites occur as a result of JAK activation; (3) Tyrosine phosphorylates STATs; (4) The receptor and STATs separate, forming homodimers or heterodimers; (5) STAT dimers attach to DNA inside the nucleus and control transcription.

keratinocyte inflammatory response induced by TNF- or IL-17A. The epidermis of psoriatic patients also had activated ERK1/2, like p38. The psoriasiform lesion caused by IMQ was reduced by the ERK inhibitor JSI287. A member of the dual-specificity phosphatase family named DUSP1/MKP-1 controls the MAPK pathway negatively. It was considerably downregulated in psoriasis patients, and DUSP1 overexpression, which targeted the ERK/Elk-1/Egr-1 signaling pathway, significantly reduced keratinocyte growth and increased death (*Gisondi et al., 2018*).

### Other signaling pathways

For epidermal hyperplasia to occur, secreted frizzled-related protein (SFRP)4, a Wnt negative regulator, is essential. DNA methylation, an epigenetic regulator, reduced SFRP4 in the skin epidermis of psoriatic patients and mice models of psoriasis. Keratinocyte hyperproliferation brought on by IL-6 was inhibited *in vitro* by SFRP4 therapy or Wnt suppression. Dermatitis resembling psoriasis brought on by IMQ was lessened by the administration of SFRP4 or by the pharmaceutical suppression of Wnt.

Hippo-Yes-associated protein (YAP) signaling has reportedly been linked to psoriasis recently. In a mouse model of psoriasis and on the skin of psoriatic patients, substantial inductions of YAP and its downstream target amphiregulin (AREG) were seen. As an oncogene, YAP encourages cell growth and prevents cell death. Additionally, by acting through an AREG-dependent mechanism, YAP silencing enhanced cell apoptosis, arrested the cell cycle, and reduced keratinocyte proliferation (*Thomas, Azad & Takwale, 2021*).

## MANAGEMENT AND TREATMENT

Various management and treatment options are available to help manage the symptoms and improve the quality of life for people with psoriasis (Table 3). The treatment approach depends on the severity of the disease, the type of psoriasis, and individual patient factors such as age, comorbidities, and preferences (*Jordan et al., 2012*).

The severity of the disease, localization (particularly in sensitive areas like the face, scalp, palms, soles, nails, and genitalia), comorbidities like psoriatic arthritis, impact on quality of life, and patient preferences are all important considerations when selecting systemic therapy for psoriasis. Achieving a PASI90 response or an absolute PASI score of 3 or less is what is considered a clinical remission;. It indicates a 90% reduction in symptoms, bringing treatment goals into line with contemporary guidelines for obtaining nearly total lesion clearance (*Hawkes, Chan & Krueger, 2017*).

However, due to the detrimental psychological effects and increased severity of the illness, people with psoriasis affecting sensitive body parts may need to modify these targets. These therapeutic objectives must be sustained throughout time, demonstrating the necessity of strict illness control (*Rendon & Schäkel, 2019*).

A patient's pleasure and quality of life must also be considered; a Dermatology Life Quality Index (DLQI) score of three or below is the goal. It is advised to alter the treatment plan to manage the patient's health better and improve their general well-being if these goals are not reached in three to four months (*Steuer et al., 2020*).

### Complications and prognosis

Psoriasis is a chronic inflammatory disorder that can affect multiple organ systems in the body and is associated with various complications. Here are some of the complications of psoriasis along with their respective research article references (*Robinson et al., 2012*).

#### *Psoriatic arthritis*

Impacts as many as 30% of people who have psoriasis, while the prevalence varies depending on how severe and long the skin condition lasts. PsA is an inflammatory arthritis that can cause irreversible joint damage and impairment if left untreated. It causes joint pain, stiffness, swelling, and enthesitis. PsA may raise the risk of metabolic syndrome, cardiovascular disease, and psychological issues including anxiety and depression in addition to its clinical manifestations (*Blackstone, Patel & Bewley, 2022*).

Treatment is customised because of its variable course, which can range from mild to severe. To address both skin and joint symptoms, biologics (such as TNF, IL-17, and IL-12/23 inhibitors) are frequently used in conjunction with disease-modifying antirheumatic drugs (DMARDs). Preventing long-term disability and improving patient

**Table 3 Management and treatment of psoriasis condition.**

| | | | |
|---|---|---|---|
| 1 | Non-pharmacologic management | Lifestyle modifications | Lifestyle modifications such as regular exercise, a healthy diet, avoiding smoking, maintaining a healthy weight, and reducing stress can help in managing psoriasis |
| | | Sun exposure | Controlled sun exposure or phototherapy can help reduce the severity of psoriasis. |
| | | Moisturizers | Using moisturizers can help in reducing the dryness and scaling of the skin in psoriasis patients |
| 2 | Pharmacologic management | Topical medications | Corticosteroids—Short-term use only; long-term corticosteroid therapy is not recommended for psoriasis due to side effects. |
| | | | Topical vitamin D analogs such as calcitriol and calcipotriene are also used for the treatment of psoriasis |
| | | | Topical retinoids such as tazarotene are used for the treatment of psoriasis |
| | | | Calcineurin Inhibitors—Tacrolimus, Pimecrolimus. |
| | | Systemic medications | Methotrexate is a commonly used systemic medication for the treatment of moderate and severe psoriasis. |
| | | | Cyclosporine is another systemic medication that is used for the treatment of moderate to extreme psoriasis. |
| | | | Acitretin is a systemic retinoid that is used for the treatment of moderate to severe psoriasis. |
| | | | Phosphodiesterase-4 Inhibitors—Apremilast, an oral small-molecule inhibitor for moderate to severe psoriasis. |
| | | | JAK Inhibitors—Deucravacitinib (TYK2 inhibitor), upadacitinib for moderate to severe psoriasis and psoriatic arthritis. |
| 3 | Biologic therapy & immunotherapy | TNF-alpha inhibitors | Certolizumab, golimumab, adalimumab, infliximab, and etanercept for moderate to severe psoriasis and psoriatic arthritis. |
| | | IL-17 inhibitors | Secukinumab, ixekizumab, brodalumab, bimekizumab for moderate to severe psoriasis. |
| | | IL-23 inhibitors | Guselkumab, tildrakizumab, risankizumab for moderate to severe psoriasis. |
| | | IL-12/23 inhibitors | Ustekinumab (for moderate to severe psoriasis and psoriatic arthritis). |
| | | IL-36 inhibitors | Spesolimab (approved for generalized pustular psoriasis). |

outcomes need early diagnosis and intervention, especially in individuals with risk factors such severe psoriasis or nail involvement (*Nast et al., 2015*; *Wahl et al., 1999*).

### Cardiovascular disease

Psoriasis is associated with an increased risk of cardiovascular disease, including heart attacks, strokes, and atherosclerosis. This is thought to be due to the chronic inflammation associated with psoriasis, which can contribute to the development and progression of cardiovascular disease (*Parisi et al., 2013*).

### Metabolic syndrome

Individuals with psoriasis have an increased risk of developing metabolic syndrome, which is a cluster of conditions that includes high blood pressure, high blood sugar, excess body fat around the waist, and abnormal cholesterol levels. Metabolic syndrome increases the risk of cardiovascular disease and type 2 diabetes (*Basavaraj et al., 2010*).

### Depression and anxiety

Psoriasis can have a significant impact on an individual's quality of life and can lead to depression and anxiety. This is thought to be due to the social stigma associated with psoriasis, as well as the physical discomfort and disfigurement caused by the condition. There is also evidence that it is not simply a secondary reactive mechanism but at least in parts independent comorbidity with *e.g.*, endogenous depression (*Fortune et al., 1997*).

The prognosis of psoriasis varies widely depending on the type and severity of the disease, as well as individual factors such as age, overall health, and lifestyle. In general, psoriasis is a chronic condition with no known cure, but it can be managed effectively with proper treatment and self-care. According to a systematic review and meta-analysis, the overall remission rate for psoriasis was 20.3%, with remission rates varying depending on the type of psoriasis (*Rigas et al., 2019*). For example, remission rates were higher for guttate psoriasis (37.4%) compared to plaque psoriasis (18.5%) (*Wang et al., 2021*).

## Comorbidities

Psoriasis is largely caused by chronic inflammation, which also increases the risk of autoimmune diseases like multiple sclerosis and inflammatory bowel disease as well as endocrine abnormalities like thyroid dysfunction. Psoriasis has been connected to chronic kidney disease, most likely because of nephrotoxicity from medications and systemic inflammation (*Fernández-Guarino et al., 2016*). Patients with psoriasis are also more likely to experience sleep problems, such as obstructive sleep apnea, which may exacerbate fatigue and negatively impact general health. These comorbidities emphasize the necessity of treating psoriasis with a multidisciplinary approach, making sure that treatment plans consider both the skin ailment and its more extensive systemic repercussions (*Boehncke & Schön, 2015*).

## Psychosocial impact

Psoriasis is accompanied by severe psychological comorbidity, which imposes a burden that goes far beyond the disease's outward clinical manifestations. Suicidal thoughts, drug abuse, and psychosocial comorbidities including anxiety and depression are all closely linked to psoriasis. According to recent studies, even while up to 98% of patients believed that their skin condition had impacted their emotional or psychological health, just 18% sought assistance. This care gap is mostly a result of inadequate detection, diagnosis, and triage, as well as a lack of knowledge about the few services that are offered (*Armstrong, Harskamp & Armstrong, 2013*).

One study investigated the impact of psoriasis on the quality of life of patients. The study found that psoriasis patients reported a reduced quality of life compared to the general population. The patients reported experiencing emotional distress, social stigmatization, and lower self-esteem due to the disease. Furthermore, the study found that patients with more severe psoriasis reported more significant negative impacts on their quality of life than those with mild or moderate psoriasis (*Koller et al., 2011*).

The patients with extreme psoriasis were more likely to experience depression than those with milder forms of the disease. The study also found that depression was more

common in patients who reported a higher degree of disability due to psoriasis. Several studies have also found that psoriasis can have a notable impact on a patient's social and occupational life. Patients may experience social isolation, discrimination, and decreased work productivity due to the disease (*Parisi et al., 2013*).

These studies highlight the significant psychosocial impact of psoriasis on patients' lives. Healthcare providers should consider the emotional and social aspects of the disease when managing psoriasis patients and provide appropriate psychological support when needed.

### Impact on quality of life

Psoriasis significantly impairs the health-related quality of life (HRQoL) of its patients. According to a National Psoriasis Foundation survey, more than 75% of patients said their illness interfered with their everyday activities. At least 20% of patients have considered suicide, indicating that emotional suffering is also prevalent (*Raho et al., 2012*). Due to the physical and psychological effects of the illness, over 60% of patients report missing an average of 26 days of work annually, which has an impact on productivity. Due to missing employment and treatment expenses, this results in financial burdens (*Strohal et al., 2015*).

Patients frequently experience stigma, humiliation, and embarrassment, as well as poor body image and bad coping strategies. These elements lead to a lower quality of life, which is further worse by actions like concealing lesions or avoiding social situations. However, psoriasis's non-contagious characteristic can improve the quality of life and ease some social discomfort (*Aurangabadkar, 2013*).

Coping strategies are essential for handling the mental and physical difficulties associated with psoriasis. The quality of life is enhanced by constructive coping mechanisms like seeking assistance, lowering stress, and keeping up excellent skincare practices. While open communication with loved ones lessens isolation, stress management practices like mindfulness and taking up a hobby can help manage symptoms (Table 4). Additionally, getting medical advice guarantees individualized care and emotional support (*Armstrong et al., 2017*).

HRQoL is also considerably improved by support groups. Patients who take part in these groups report feeling less depressed and anxious, knowing more about their illnesses, and following their treatment regimens more closely. These communities offer emotional support and a feeling of belonging, which enhances general well-being (*Svanström, Lonne-Rahm & Nordlind, 2019*).

## FUTURE DIRECTION FOR TREATING PSORIASIS CONDITION

Several promising treatments that address the intricate and multifaceted nature of psoriasis have been made possible by recent developments in the field. Targeted therapies have shown considerable clinical success, especially in inhibiting the IL-23 and IL-17 pathways, even though psoriasis is still incurable. Some patients report long-lasting anti-psoriatic effects even after stopping treatment, suggesting that these medicines can modify the condition over the long term. Nevertheless, more investigation is required to comprehend why, despite their early effectiveness, some biologics only produce transient clinical

**Table 4 Positive and negative coping strategies associated with better QoL.**

| Positive coping mechanisms | Negative coping mechanisms |
|---|---|
| Education and knowledge | Social isolation and withdrawal |
| Seeking support | Substance abuse |
| Self-care practices | Negative self-talk |
| Open communication | Excessive use of topical treatments |
| Seeking professional help | Avoidance of treatment |

responses, frequently requiring a change to alternative therapy and an examination of the short medication duration of some treatments. Biologics that target TNF-alpha, IL-23, and IL-17 have revolutionized the treatment of psoriasis in recent years. TNF-alpha inhibitors like infliximab and adalimumab have shown promise in treating moderate to severe forms of the condition. More targeted alternatives are now available thanks to the recent approval of JAK inhibitors and IL-36 inhibitors (for pustular psoriasis). Particularly for patients who want non-injectable treatments, next-generation small molecules, including RORt inhibitors, which are presently being studied, show potential as oral substitutes. However, because of their affordability and accessibility, broader-acting medications like methotrexate and cyclosporine continue to be crucial in many therapeutic settings, especially in areas where sophisticated biologics are not easily accessible. As we move forward, assigning patients to the best therapies and making sure that treatment is durable over the long term will be a major problem. It is crucial to tailor treatment according to cytokine profiles, genetic predisposition, and other variables affecting treatment response. Improving treatment approaches to control symptoms, lower relapse rates, and enhance patients' quality of life will continue to be a top objective in the treatment of psoriasis as research advances (*Navarini et al., 2017*).

## CONCLUSION

Psoriasis is a chronic autoimmune skin condition that causes red, scaly areas that cause pain, discomfort, and social stigmatization. It affects 2 to 3 percent of people worldwide. Numerous varieties of psoriasis, including nail, plaque, guttate, inverse, pustular, and erythrodermic, result from its intricate aetiology, which is a result of the interaction of immunological, environmental, and genetic variables. The clinical presentation is the primary basis for diagnosis; imaging, blood testing, and biopsy are also used. Psoriasis development is significantly regulated by keratinocytes, which are in turn impacted by genetic factors, cytokines, metabolism, and cell signaling. Different treatments, such as topical, systemic, biologic, pharmacologic, and non-pharmacologic therapy, try to control symptoms even if there is no cure. Patients' quality of life is greatly impacted by psoriasis, which frequently results in mental health problems including anxiety and sadness. Support networks and coping strategies are essential for handling emotional stress. The main objective of treatment is to achieve clinical remission based on a PASI90 response or an absolute PASI score of three. The relevance of dysbiosis is highlighted by microbiome research, which connects microbial imbalances to inflammatory reactions and

disturbances of the skin barrier. An understanding of these complexities guides targeted therapies. There is variation in worldwide prevalence; greater lifetime prevalence rates are found in Australia, Norway, and Israel, among other nations. Psoriasis was expected to affect 29.5 million adults globally in 2017, highlighting the disease's enormous impact on people worldwide. Accurate diagnosis and interdisciplinary care help patients reduce symptoms and improve their quality of life in general.

## ACKNOWLEDGEMENTS

Sai Chakith M R, Sushma Pradeep, Manu Gangadhar, Chaithra Maheshwari N, Shuaib Pasha, Satish Allur Mallanna, Chandan Shivamallu acknowledge the institutional infrastructure support including access to research laboratories, equipment, and administrative assistance throughout the course of the study offered by the JSS Academy of Higher Education and Research (JSSAHER), Mysuru, India. SPK is grateful to the Director, Amrita Vishwa Vidyapeetham, Mysuru campus, for infrastructure support.

### Funding

The authors received no funding for this work. Sushma Pradeep received a Senior Research Fellowship from the Indian Council for Medical Research. The funders had no role in study design, data collection and analysis, decision to publish, or preparation of the manuscript.

### Grant Disclosures

The following grant information was disclosed by the authors:
Indian Council for Medical Research.

### Competing Interests

The authors declare that they have no competing interests.

### Author Contributions

- Sai Chakith M. R. conceived and designed the experiments, performed the experiments, prepared figures and/or tables, and approved the final draft.
- Sushma Pradeep conceived and designed the experiments, performed the experiments, prepared figures and/or tables, authored or reviewed drafts of the article, and approved the final draft.
- Manu Gangadhar performed the experiments, authored or reviewed drafts of the article, and approved the final draft.
- Chaithra Maheshwari N. conceived and designed the experiments, prepared figures and/or tables, and approved the final draft.
- Shuaib Pasha performed the experiments, analyzed the data, authored or reviewed drafts of the article, and approved the final draft.
- Shiva Prasad Kollur analyzed the data, authored or reviewed drafts of the article, and approved the final draft.

- Nagashree S. analyzed the data, authored or reviewed drafts of the article, and approved the final draft.
- Chandan Shivamallu analyzed the data, authored or reviewed drafts of the article, and approved the final draft.
- Satish Allur Mallanna analyzed the data, authored or reviewed drafts of the article, and approved the final draft.

## Data Availability

This is a literature review.

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
