# Peer review of "Advancements in understanding and treating psoriasis: a comprehensive review of pathophysiology, diagnosis, and therapeutic approaches"

_PeerJ, doi:10.7717/peerj.19325_

## Round 0.1 · original submission · Major Revisions

Please address the comments of all 3 reviewers. We look forward to your revision

Reviewer 1 ·

Basic reporting

This article mostly favorably reviews psoriasis, however, some information is defected.

Experimental design

Mostly the composition of the review is fine.

Validity of the findings

Mostly valid.

Additional comments

Some therapeutic modalities are deleted in Table 3, and should be incorporated:
Section 6 (immunotherapy) must be included in section 3.
Section 4: tapinarof (AhR agonist) should be added.
Section 5: apremilast deucravacitinib, upadacitinib should be added.
Figure 1: The pictures do not represent the features of individual types of psoriasis.

·

Basic reporting

The article is written in English language but needs corrections. Maybe the authors can ask help by a language expert.
In the text on line 243 the reference needs correction, the writing year cannot be 2027.
Οn line 515 and 591 there are brackets with a number, if this corresponds to a reference it should be written differently, or deleted.
The table 3 needs corrections: the drugs bimekizumab, guselkumab, ustekinumab are not immunotherapy are biologic therapy.
The theme of the article is very interesting, authors provide updated epidimiologic characteristics of psoriasis, and give interesting details about the pathophysiology with figures.
In “management and treatment” section the therapeutic target in psoriasis nowadays is EASI 90 and is important that the authors mentioned it.
The author’s introduction is satisfied.

Experimental design

The article content conforms to journal specifications.
Methods described with sufficient information to be reproducible by another investigator.
The review is organized logically into paragraphs.

Validity of the findings

The paper does not provide novelty.
The field of psoriasis has been reviewed recently but is always very interesting and challenging theme.
Conclusions are well stated, linked to article presentation, and identify unresolved questions. Discusses the topic of biomarkers that is a future directions in the disease

Additional comments

Needs correction for publishing.

Reviewer 3 ·

Basic reporting

Dear Authors, thank You very much for this intersting overview to summarize nowadays knowledge of this chronic inflammatory disease. It seems that You made a huge effort to include all detailed research going on for understanding of molecular mechanisms of this disease and combine it with common knowledge which might be very challenging, especially for Your intended readers; also there are numerous repetitions which give the impression of several independent writers.
May I address some points:
• Line 50: the age of onset is mainly recognized with two main peaks as early onset and late onset but can also be found in very young children as well
• Line 60: this subdivisions are irritating: there are topical and systemic medications, and systemics can be oral or subcutaneous or intravenously, biologics are not the only subcutaneous meds (e.g. MTX)
• Line 95: please mention a ratio men:women because this is not too impressive as it might be anticipated from Your sentence and more, found to be quite equal
• Line 100-121: perhaps You could shorten this for better overview and mention rough trends instead of detailed numbers, refer also to the WHO global report on psoriasis (Global report on psoriasis (who.int))
• Line 125: I would prefer not the term “threat” but risk
• Line 138: trigger factors do not relate to develop psoriasis (because it it already genetically and multifactoral) but to express it
• Line 144-280: there are numerous repetitions, sometimes two sentences right after each other (e.g. line 205-210 and many more) and perhaps it is better to give a shortened overview instead of repeating all bullet points in every subchapter
• Line 147: in typical locations
• Line 162: metabolic syndrome, psoriatic arthritis or heart disease are no side effects but comorbidities
• Line 164: if You mention several types of cancer, for this You should name a reliable source, because this is not a real higher risk of psoriasis
• Line 169: psoriatic arthritis is not a uniform involvement of joints and You should mention that we have knowledge of distinct types of psoriatic arthritis which react very differently to treatment
• Line 189: psoriasis is multifactorial and not related to only one gene which would be anticipated from Your sentence
• Line 192: cytokines are better described as molecules or factors than chemicals
• Line 194: perhaps You should add that the interleukins promote the inflammatory cascade
• Line 202: guttate psoriasis is not exclusively in kids and teenagers but also in adults
• Line 219: those areas can be, but are not always all affected
• Line 252: flares of pustular psoriasis are unpredictable in most cases
• Line 255: would not use the term “easy” because pustular psoriasis is one of the most challenging forms of psoriasis to treat and very limited treatment choices especially for GPP
• Line331 ff: x-rays and MRT can only detect lesions which are already developed and nowadays high resolution ultrasound of joints and cartilage detect early signals for inflammation better, therefore x-rays lost their significance for detections and control of inflammation as we try to find psoriatic arthritis early for treating specifically this inflammation and to prevent damage
• Line 334: dermoscopy does not play a significant role in diagnosing psoriasis, perhaps in distinguishing between differential diagnosis, better is clinical appearance and histology gold standard
• Line 337: teledermatology is good for follow-ups and first anamnestic contact, but not for the first complete thorough investigation, because it is important to see the entire skin including genital areas which are often involved, as well as to assess stigmatizing cofactors which will not be properly addressed in teledermatology, see also existing guidelines for teledermatology (AAD, German, British)
• Line 354, 360, fig 3: You use the term hyperplasia; this is not correct, it is a epidermopoesis with hypertrophy und hyperproliferation
• Line 355: it is more an elongation of existing vessels than a neovascularization
• Line 370: it is not the safety of biologics which initiates more research and Your sentence might mislead, biologics are very specific in their class and each group has a different target and effect so that they all differ very significantly in side effects, in treatment conduct, in long-term effects and super- or non-responders
• Line 379: this is not the rule, psoriatic lesions do not persist only due to microbio changes, it is more complex
• Line 385-540: numerous repetitions together with a very detailed overview over signalling pathways; You mention in the beginning that this manuscript intends to give an overview for advocates, lawmakers and healthcare regulators: this audience might be challenged with this chapter, perhaps mention that there is a lot research going on to find more evidence for the role and driving factors and shorten to the factors which already lead to development of approved medications like JAK / TYK inhibitors
• Line 428: IL-22 promotes not prevents psoriatic inflammation and with its stimulation of antimicrobial peptides explains why psoriatic lesions do not tend to superinfect like lesions of atopic dermatitis
• Line 556 ff.: repetition
• Line 562 ff: already mentioned earlier and repeats very roughly what You described previously, might be better to describe the comorbidities in the chapter 1, especially 3.1.1 needs more attention
• Line 581: there is also evidence that it is not simply a secondary reactive mechanism but at least in parts independent comorbidity with e.g. endogenous depression
• Line 589: we do not see in general a susceptibility to superinfections in psoriasis and as You mentioned before there are cytokine and microbiol factors supporting evidence for a barrier function different from e.g. atopic dermatitis where filaggrin mutation and a different microbiome shows more superinfection and different staphylococcal susceptibility
• Line 597: another repetition
• Line 604 ff: see above, chapters can be summarized together
• Line 676ff: this overview adds no further support, too superficial for the range of possibilities, please refer to existing guidelines for treatments, line 679 lacks TNFalpha, lacks JAK approvals and IL-36 approval for pustular psoriasis, there is no clear distinction between former conservative approaches and modern biologics and small molecules
• Line 697 ff: perhaps prefer stigmatization instead of shame, for conclusion too much weight on microbial significance, for nail changes and assessment of the NAPSI score You do not necessarily need dermoscopy
• Table 1: pustules as sterile, guttate psoriasis has also scaling
• Table 3: please do not mention brand names (bimzelx)
• Table 3: the headers of the lines are not clear, overlaps between 2-6 (all pharmacological), no clear topical and systemic discrimination, topicals include also dithranol, salicylic acid, clacineurininhibitors,
ustekinumab should be mentioned as IL12/23 inhibitor, distinction non-bios/bios, corticosteroids lost its significance in treatment and are only recommended for short term intervention, not as long-term treatments, please refer to existing guidelines, TNFalphas are missing (certolizumab, golimumab), the groups of interleukines are not complete (IL17 has also bimekizumab and brodalumab, IL23 has risankizumab and tildrakizumab, Il-36 with spesolimab), JAKS are missing completely, mention perhaps also different approvals for psoriasis and psoriatic arthritis, too much details for some bios (guselkumab, ustekinumab) and some not mentioned therefore deysbalance which should not be intended
• Table 4: do You have a reference for negative coping “excessive use of topical treatments”
• Fig 1: these illustrations are not helping in imaging the distinct features

I hope that this review supports You in work on Your paper.
Best regards.

Experimental design

see above

Validity of the findings

see above

Additional comments

Perhaps You should consider to write two papers, one on common knowledge and one on research to pathophysiology and mechanisms of psoriasis.

---

## Round 0.2 · Minor Revisions

Almost there, Dr Shivamallu - just a few relatively minor issues to address.
1) One line 61-62 - please use your own wording.rather than those of the reviewer.
2) Remove repetitions (see: line 97-100 repeats 102-103, 208 doubles 211, 235 doubles 238.
3) In line 321 there is the sentence “… as You said…”, where does this refer to?
4) Line 325 references missing.
5) Line 390 states “…data were collected…”, provide context.
6) Chapter 2.3.2. misses references
7) Line 531-534 mentions the EASI. This related to atopic dermatitis not psoriasis; Please address and supply supporting reference.
8) Chapter 3.1.5 , line 575. Comments on skin barrier function and infections are not the main problem of psoriasis (unlike for atopic dermatitis). Revise working here.
9) Repeition: Chapter 3.2 doubles 3.1. Please address.
10) Table 3 has still overlaps topicals and systemics.

Reviewer 1 ·

Basic reporting

Well organized.

Experimental design

Favorable design

Validity of the findings

Mostly Valid

Additional comments

No comments

·

Basic reporting

No comment.

Experimental design

Article content is within the aims and scope of the journal and article type. Methods are described with sufficient details and all recommendations have been taken into account and modified accordingly so that the article is publishable.

Validity of the findings

Conclusion is well stated, linked to original research question and limited to supporting results. Is connected to the original question investigated, and is well developed after reviewers suggestions.

Additional comments

No additional comments.

Reviewer 3 ·

Basic reporting

no comments

Experimental design

Throughout the manuscript there are some references missing. see comments below

Validity of the findings

no comments

Additional comments

Dear Authors,
thank You very much for Your dedicated work on this topic. It is very challenging to give a good overview for every aspect of this disease and I appreciate Your efforts to improve the manuscript.
May I mention some points:
1. Some of my comments were copied into the manuscript without further work, see line 61-62, this should be a hint to choose an own wording.
2. There are still several doubles and repetitions throughout, e.g. line 97-100 repeats 102-103, 208 doubles 211, 235 doubles 238,…, as previously mentioned, please review Your manuscript for these repetitions.
3. There are several sentences with”… the authors noted…” (line 252,270,272,…) could You perhaps find some references to support Your own impressions.
4. In line 321 there is the sentence “… as You said…”, where does this refer to?
5. Line 325 ff there are references missing.
6. Line 390 states “…data were collected…”, this is without context?
7. Chapter 2.3.2. misses references entirely for such statements.
8. I am still irritated about the target audience, You give repeated details about molecular pathomechanisms which are in parts only listings and sequences, could You perhaps find a more compact way to describe it?
9. Line 531-534 mentions the EASI, this is a tool for atopic dermatitis and has nothing to do with this manuscript and psoriasis; could it perhaps be a copy paste problem and do You need to reference before?
10. Chapter 3.1.5 , line 575 ff. as mentioned already, the skin barrier function and infections are not the main problem of psoriasis, this is different from atopic dermatitis and sometimes confused by AI or other tools; please could You do this chapter more precise with the corresponding references.
11. Chapter 3.2 doubles 3.1., please shorten the doubles.
12. Table 3 has still overlaps of topicals and systemics and the listing, perhaps You should use the categorizations of national and international guidelines.
I hope that this support Your efforts for a good manuscript.
Sincerely.

---

## Round 0.3 · accepted · Accept

Thank you for resolving these remaining minor issues, which has helped to streamline the manuscript and mke it more readable for the reader.